# Psychosocial experiences of adolescents with tuberculosis in Cape Town

**Dillon T. Wademan[1‡]\*, Mfundo Mlomzale[1‡], Arlene J. Marthinus[1], Stephanie Jacobs[1], Khanyisa Mcimeli[1], Klassina Zimri[1], James A. Seddon[1,2], Graeme Hoddinott[1,3]**

1 Desmond Tutu TB Centre, Department of Paediatrics and Child Health, Faculty of Medicine and Health Sciences, Stellenbosch University, Cape Town, South Africa, 2 Department of Infectious Disease, Imperial College London, London, United Kingdom, 3 School of Public Health, Faculty of Medicine and Health, The University of Sydney, Sydney, Australia

‡ DTW and MM contributed equally as first authors.
* dtwademan@sun.ac.za

## Abstract

Adolescents (10-19-years-old) account for almost 10% of the annual global tuberculosis (TB) incidence. Adolescents' experiences of TB care, TB stigma, and the consequences of TB for their relationships, schooling, and mental health are different, and often more severe, compared to younger children and adults. How TB impacts the lives of adolescents is not well described or understood. We aimed to locate adolescents' experiences of TB relative to their psychosocial contexts, describe the impact of TB on adolescents' wellbeing, and describe how TB and its treatment affects their socio-familial contexts. Teen TB was a prospective observational cohort study which recruited 50 adolescents with newly diagnosed TB disease (including both multidrug-resistant TB and drug-susceptible TB) in Cape Town, South Africa. A nested sub-sample of 20 adolescents were purposively sampled for longitudinal qualitative data collection. Nineteen participants completed all qualitative data collection activities between December 2020 and September 2021. Adolescents described their communities as undesirable places to live—rife with violence, poverty, and unemployment. The negative experiences of living in these conditions were exacerbated by TB episodes among adolescents or within their households. TB and its treatment disrupted adolescents' socio-familial connections; many participants described losing friendships and attachment to family members as people reacted negatively to their TB diagnosis. TB, inclusive of the experience of disease, diagnosis and treatment also negatively impacted adolescents' mental health. Participants reported feeling depressed, despondent, and at times suicidal. TB also disrupted adolescents' schooling and employment opportunities as adolescents were absent from school and college for substantial periods of time. Our findings confirm that adolescents' psychosocial experiences of TB are often highly negative, compounding underlying vulnerability. Future research should prioritize exploring the potential of social protection programmes providing adolescents and their families with psychosocial and economic support.

**Data Availability Statement:** The datasets generated and analyzed during the current study are not publicly available. The consent forms and study protocol approved by Stellenbosch

University Health Research Ethics Committee preclude publicly sharing the data. However, the data are available upon reasonable request. Requests can be directed to the Health Research Ethics Committee at Stellenbosch University (ethics@sun.ac.za).

**Funding:** This work was supported by J.S. who received a Clinician Scientist Fellowship from the UK Medical Research Council (MRC) and the UK Department for International Development (DFID) through the MRC/DFID Concordat agreement, which sponsored this work. G.H. is supported by funding from financial assistance of the European Union (Grant no. DCI-PANAF/2020/420-028), through the African Research Initiative for Scientific Excellence (ARISE), pilot programme. ARISE is implemented by the African Academy of Sciences with support from the European Commission and the African Union Commission. The contents of this document are the sole responsibility of the author(s) and can under no circumstances be regarded as reflecting the position of the European Union, the African Academy of Sciences, and the African Union Commission.

**Competing interests:** The authors have declared that no competing interest exist.

## Background

Adolescents (10-19-years-old) account for almost 10% of the annual global tuberculosis (TB) incidence [1, 2]. Adolescents are an orphaned population in TB surveillance programmes, due to data being reported for 'children' (aged 0-14-years-old) and 'adults' (aged ≥15-years-old) [1, 3]. In 2020, an estimated 1.09 million children and young adolescents (aged 0-15-years-old) developed TB, with only 400,000 recorded cases, indicating a substantial diagnosis and reporting gap [4].

However, adolescents are increasingly being recognised as an important group for TB interventions, treatment, and epidemic control because of their biosocial vulnerability to *M. tuberculosis* infection and TB disease progression [5]. Adolescents predominantly present with infectious forms of TB, as mycobacteria are commonly present in the sputum. Diagnosis, is therefore, commonly more straightforward than for younger children. However, treatment outcomes are worse than in children and adults [5]. Reasons for these worse outcomes are unclear but may be due to physiological and psychological transitions that occur during this life stage [6, 7]. Yet, reaching adolescents and providing them with appropriate services remains a challenge in South Africa [8, 9]. The complex ways in which biological vulnerability and systemic restraints intertwine with behavioural changes and social development complicate treatment and care provision for adolescents [10]. More broadly, the long-term impact of TB on adolescents' lives cuts across all domains of life, including the physiological, the social, the psychological, and potentially even educational which may in turn negatively impact their future [11]. These impacts are often linked to treatment processes involved in TB care (and comorbidities) [2].

Typically, adolescents diagnosed with drug-susceptible TB are treated for 6 months, while those diagnosed with multidrug-resistant (MDR-)TB are treated for 9–18 months [3]. MDR-TB is defined as disease caused by *M. tuberculosis* resistant to at least isoniazid and rifampicin, and treatment has historically included drugs with substantial adverse effects [12, 13]. In many places, prolonged home isolation or hospitalisation remain routine practice for adolescents diagnosed with drug-resistant forms of TB, leading to major disruptions in education, social relationships and decline in general well-being [14, 15]. The introduction of new and repurposed drugs, shorter regimens, better formulations, and injectable-free all-oral regimens for children are promising innovations [16–19]. However, some of these innovations are not yet widely available and many available drugs remain poorly palatable and are disruptive to everyday life [20, 21].

How adolescent's healthcare-seeking behaviours and overall wellbeing are affected by their social and familiar context is not well described or understood. 'Wellbeing' has long been a subject of contestation but its underlying motif is that people, including adolescents, are permitted the ability to flourish in every domain of their lives [22, 23]. A recently published review of the impact of TB on the wellbeing of adolescents suggested five domains being crucial for the successful transition from childhood to adulthood: (1) good health; (2) connectedness and contribution to society; (3) safety and a supportive environment; (4) learning, competence, education, skills, and employability; and (5) agency and resilience [11].

In this paper we aim to (a) locate adolescents' experiences of TB relative to their economic and health contexts (b) describe the impact of TB disease (including individuals with both drug-susceptible and MDR-TB) on adolescents' psychosocial, socio-familial, and school / employment wellbeing and, (c) consider adolescents' agency and resilience.

## Methods

### Setting

Teen TB was a prospective observational cohort study which recruited 50 adolescents (10-19-years-old) with newly diagnosed microbiologically confirmed pulmonary TB disease (including both MDR-TB and drug-susceptibe TB), with or without HIV coinfection and who were within the first 14 days of treatment. Participants were followed over 12-months [24]. Teen TB was based in the Cape Metropolitan District of the Western Cape Province, South Africa. This is an urban/peri-urban setting of ~1000sqm/~2500km2, with ~4,5 million residents, >100 primary care health facilities and one of the highest TB burdens in the world, with ~28,776 case notifications for the 2022–2023 year [25]. In 2013, adolescents accounted for 7.5% of all new TB notifications in the Western Cape [26].

### Sampling and recruitment

Participants to the parent Teen TB study were recruited from the Tygerberg, Mitchell's Plain, and Khayelitsha subdistricts of the City of Cape Town between December 2020 and September 2021. A nested sub-sample of 20 adolescents were purposively sampled for diversity and richness. The initial invitation to the qualitative sub-study was done by the clinical research staff and study nurse/counsellors. Thereafter two graduate socio-behavioural science researchers contacted the participant (and legal guardian) to set up a formal introduction to the qualitative sub-study.

### Data collection

Nineteen of the 20 participants recruited to the qualitative sub-study completed all study activities. One participant was lost to attrition and was withdrawn from the study. At each interaction, participants were interviewed using an activity-based, semi-structured interview schedule. The graduate researchers (KM and SJ) received study-specific training on how to execute the participatory research activities to achieve implementation uniformity. Topic areas covered over the four interactions include familial/peer social contexts, experiences of TB disease, treatment and care, sexual relationships, comorbidities, substance use, education, and overall quality of life. Each interaction ranged from 15 to 90 minutes and was conducted in the participants' preferred language (English, Afrikaans, or Xhosa). All interactions were recorded, and photographs were taken of completed participatory research activities. After each interaction, the graduate researchers completed a detailed case description. Case descriptions were shared with senior researchers (DW and GH) for review and input—which, in turn, informed future interactions with participants [27].

### Data analysis

Qualitative analysis compromised a multi-step thematic analysis. Moscibrodzki et al.'s [11] framework of adolescent well-being guided the analytic process. After all interactions, participant's case descriptions for each interaction were collated into single case files. Researchers (DW, AM, and MM) identified quotes from each recorded interaction and added these to the case files to illustrate key narratives relating to each of five domains of adolescents' wellbeing. Across and within-case analysis was conducted to identify experiences particular to each participant, as well as important patterns and experiences across the data set [28]. The interpretation of data was checked by GH and JS for analytic uniformity.

### Ethics considerations

The study received ethical approval from the Stellenbosch University Health Research Ethics Committee (N19/10/148). Adolescents younger than 18-years-old provided written assent and their caregivers written consent. Adolescents aged 18-19-years provided their own consent. Confidentiality was secured by de-identifying participants at the point of data capturing and assigning pseudonyms in case files and data transcription.

## Findings

### Participant characteristics

We conducted 47 interviews with 19 adolescents (9 girls and young women; 10 boys and young men). Adolescents included nine Xhosa-speakers, eight Afrikaans-speakers and two English-speakers. Adolescents were at various stages in their treatment journeys when interviewed, with some having recently completed their treatment, while others were still on treatment. Twelve adolescents had been diagnosed with drug-susceptible TB, while seven had MDR-TB. One participant was living with HIV. See Table 1 for a summary of findings.

### Experiences of TB in environmental and health contexts

Adolescents generally described their communities as undesirable places to live, not conducive to healthy living. One participant said that a doctor encouraged her entire family to relocate to another community because their previous community was 'dirty'. Dirt was closely linked with impoverishment and poor housing. Some participants associated spending time with people they considered living in dirty areas as increasing their risk of infection and/or aggravating their disease. A 17-year-old man with drug-susceptible TB said:

> *"There's nothing wrong [with being around other people] it's just that you don't know what other sicknesses they might have. [. . .] I am worried that my health will worsen [by spending time with people]."*

Although food and nourishment were widely recognised as an important route to good health with one young man who had recently turned 20-years-old and experienced MDR-TB saying: *"I try to have an orange or apple on a daily basis or have good food so that I can feel better"*. Many other participants described food scarcity being a major problem to achieving optimal health and adhering to their treatment. For example, a 17-year-old boy with drug-susceptible TB said: *"because there was no food, I would sometimes forget to take my medication in the morning."* Some participants reported forcing themselves to take their medication on an empty stomach, even if it resulted in worse side-effects. For example, a young woman who had recently turned 20-years-old and experienced MDR-TB said:

> "*I take pills even if there is no food [and] then I get a slight headache. Even if there is no food, I have to take my pills."*

Adolescents in our study reported experiencing dizziness, vomiting, nausea, and fatigue. These experiences negatively impacted their everyday routines. For example, an 18-year-old man with MDR-TB said:

> "*The treatment made me dizzy. I become nauseas and felt like vomiting, and I felt like skipping [treatment] that day. The nausea and dizziness would normally take an hour and I could not do anything in that period."*

**Table 1. Summary of findings.**

| Domain of Well-Being | Ways in which TB impacts well-Being | Participants quotes |
|---|---|---|
| Experiences of TB in environmental and health contexts | Healthy living | "There's nothing wrong [with being around other people] it's just that you don't know what other sicknesses they might have. [. . .] I am worried that my health will worsen [by spending time with people]" (17-year-old boy). |
| Impact on psychosocial wellbeing | Physical changes | "I just want to like get back to me and do what I used to love, like sports and being fit. I used to love fitness and I used to work out a lot. Then [when I got TB] I got thin and stuff" (19-year-old man). |
|  | Mental health | "This TB thing is not just a physical challenge, it's a mental challenge as well" (19-year-old man)<br>"We are all going through a lot, but I don't know how to handle this [TB treatment] [. . .]like there's times when I ask my creator just to take me away man. If my life is like this just take me away man [. . .] I don't want to be here anymore" (19-year-old man).<br>"I could do nothing. I took a few spoons of food at night then vomited. I laid alone in bed [. . .] I had no connection to another person. So, my relationship with my daughter changed due to TB. It was shocking to be this way" (19-year-old woman).<br>"Physically, I've lost a lot of weight, which has put a lot of pressure on my self-esteem; before I had MDR-TB, I was confident and wore short pants and a vest, but now I'm usually wearing [long sleeve] clothes and don't feel so well" (20-year-old man).<br>"I spend time with the rest of my loved ones" (17-year-old boy). |
| Impact on community and familial wellbeing | Ability to feel connected to family and friends | "My life has changed a lot, because I had to give up many things, like drinking alcohol. It was during the festive season, and I couldn't go out with my friends, I was always indoors. My friends went to the Eastern Cape, and I was alone, always wearing a mask because I didn't want to infect others" (15-year-old boy). |
|  | Connectedness with romantic partners | "And that time I had to stay away from girls for 6 months. Like I couldn't be around girls, like contact or kiss or whatever and that was also a thing" (19-year-old man).<br>"It affected my relationship with my boyfriend because we could not see each other. We only communicated through a cell phone. I couldn't go to him when I was sick" (A 15-year-old girl). |
|  | Impact of TB on their peer relationships | "They are my three friends, and when I first had TB, they were the ones who woke me up in the morning to take the pills [. . .] because I have to take the tablets at a specific time, and they kept reminding me" (20-year-old man) |
|  | Experienced stigma | "My aunt would go around and clean every surface [after I had been there]. She behaved like I brought this thing [TB] into the house. She made everyone in the house paranoid, including me. She even isolated my cutlery [. . .] I was in state, I already felt like I'm endangering my family" (19-year-old man). |
|  | Anticipated stigma | "Going to infect the children [and] they must stay away from me" (12-year-old girl). |
|  | Internalised (or self) stigma | "My girlfriend is not suspicious; I didn't tell her about my TB because she'll never want to see me again. She won't see it in the same way that others do. [. . .] She'll leave, I know her. [. . .] I don't want my friends to know, all of them because they are silly. That's why I just stayed at home. [. . .] I only told my best friend that I have chest problems and did not tell him about TB because he was going to tell them" (17-year-old boy) |
| Impact on schooling and employment wellbeing | Managing treatment and schoolwork simultaneously | "At school, they don't care [. . .]. They did not send work for a long time. They failed to provide the homework so that I could study for the exams. They told me at school that I would have to repeat Grade 9" (15-year-old girl).<br>"Vomiting [and] unable to eat, sleep, or do anything. So, I was unable to pursue my dreams [of] becoming a doctor" (19-year-old woman). |
| Impact on agency and resilience | Limited agency due to being diagnosed with TB | "My aunts and uncles were visiting one weekend, and my mother told them about my TB diagnosis because she was afraid that they will discriminate me" (15-year-old girl). |
|  | Resilience in the face of TB and its treatment | "When I first found out that I have TB, I was angry. Because I remembered that my dad died of TB, and I thought I was going to die as well. But now I am better because nurses motivated me to take my medication" (15-year-old girl).<br>"For the next 6 month [my goal] is to get a laptop and start applying for a bursary at the end of my grade 11 year. [. . .]. After school I would like to study business and then establish myself and after that I would like to find a romantic partner." (16-year-old boy). |

Participants with MDR-TB and/or living with HIV found their treatment journeys especially difficult. Living with HIV and being diagnosed with, and on treatment for TB, magnified overlapping negative treatment experiences, including side effects, pill burden, and stigma. For the one participant living with HIV, a recently 20-year-old young man, being diagnosed with drug-susceptible TB was overwhelming, and led him to question whether he would survive:

*"I stopped ARVs in 2019, I last took them in December [. . .] then I started again in May 2021 [. . .] I lost hope because I was doing grade 12 when I got sick and then I heard that I have TB. I didn't know if I would be given pills and I would be alright, I thought this was my death."*

## Impact on psychosocial wellbeing

Experiencing TB negatively impacted participants' ability to participate in activities that they had previously enjoyed. For example, a 19-year-old man diagnosed with drug-susceptible TB described the physical changes which inhibited his ability to exercise following his TB diagnosis:

*"I just want to like get back to me and do what I used to love, like sports and being fit. I used to love fitness and I used to work out a lot. Then [when I got TB] I got thin and stuff."*

These negative impacts also extended to his mental health: "*this TB thing is not just a physical challenge, it's a mental challenge as well.*" Participants described feeling depressed, hopeless and sometimes suicidal. The young man went on to describe how his TB diagnosis and treatment journey resulted in him questioning whether fighting for his life was worthwhile:

*"We are all going through a lot but I don't know how to handle this [TB treatment] [. . .] like there's times when I ask my creator just to take me away man. If my life is like this just take me away [. . .] I don't want to be here anymore."*

TB also negatively impacts adolescents' self-image and identities. A 19-year-old, young mother, with drug-susceptible TB, complained that she could not look after her daughter anymore. The loss of identity as a mother was difficult, leaving her feeling more isolated and alone,

*"I could do nothing. I took a few spoons of food at night then vomited. I laid alone in bed [. . .] I had no connection to another person. So, my relationship with my daughter changed due to TB. It was shocking to be this way."*

Similarly, for other participants, TB, its treatment, and correlated changes to their bodies and everyday activities resulted in a loss in self-esteem. A 20-year-old man with MDR-TB described the changes he experienced to his body and subsequently, persona.

*"Physically, I've lost a lot of weight, which has put a lot of pressure on my self-esteem; before I had MDR-TB, I was confident and wore short pants and a vest, but now I'm usually wearing [long sleeve] clothes and don't feel so well."*

Adolescents often connected changes to their appearance, and fears about how their peers perceive them, to their self-confidence, and how they interacted with others. A 17-year-old boy conceded that TB affected how *"I spend time with the rest of my loved ones."*

## Impact on community and familial wellbeing

TB and its treatment disrupted adolescents' ability to feel connected to family and friends and may impede their ability to contribute to society. The most immediate impact a TB diagnosis had on our participants was on their social lives. Adolescents in this study reported isolating themselves from others to prevent onward transmission of TB. A 15-year-old adolescent boy being treated for MDR-TB complained:

> *"My life has changed a lot, because I had to give up many things, like drinking alcohol. It was during the festive season, and I couldn't go out with my friends, I was always indoors. My friends went to the Eastern Cape, and I was alone, always wearing a mask because I didn't want to infect others."*

For other participants, TB resulted in a loss of connectedness with romantic partners. A 19-year-old man reported being frustrated by being unable to engage in romantic, physical, relationships while having TB:

> *"And that time I had to stay away from girls for 6 months. Like I couldn't be around girls, like contact or kiss or whatever and that was also a thing."*

A 15-year-old girl participant reported using social media platforms and digital technologies to help overcome the loss in connectedness, whilst simultaneously preventing spread of TB to their partners:

> *"It affected my relationship with my boyfriend because we could not see each other. We only communicated through a cell phone. I couldn't go to him when I was sick."*

Participants noted varying experiences on the impact of TB on their peer relationships. Some reported that they received support from their peers after disclosing that they have TB, reinforcing their connectedness within their peer group. For example, a 20-year-old man explained how three friends helped him adhere to his MDR-TB treatment:

> *"They are my three friends, and when I first had TB, they were the ones who woke me up in the morning to take the pills [. . .] because I have to take the tablets at a specific time, and they kept reminding me."*

For other participants disclosing their TB diagnosis, either purposefully or inadvertently, resulted in stigmatisation. Stigma negatively impacted adolescents' relationships with friends, family, and intimate partners, hindered health care access, and even interrupted treatment adherence. Some participants experienced enacted stigma and were actively shunned from social interactions or subject to emotional abuse. Participants also anticipated being stigmatized by community members or health workers in their pursuit of healthcare. Finally, some adolescents internalized pejorative narratives about themselves because of their TB diagnosis and related care needs.

A 19-year-old man participant described how he was stigmatized by a household member and how stigmatizing discourse was subsequently taken up by other family members, resulting in him thinking pejoratively about himself:

> *"My aunt would go around and clean every surface [after I had been there]. She behaved like I brought this thing [TB] into the house. She made everyone in the house paranoid, including*

*me. She even isolated my cutlery [. . .] I was in state, I already felt like I'm endangering my family."*

Other participants had similar experiences within their friendship groups. A 12-year-old girl complained about how her friend, whom she trusted with her TB diagnosis, warned everyone at school, saying that she is *"going to infect the children [and] they must stay away from me."* For this reason, participants reported being hesitant or afraid to disclose their TB diagnosis to their friends or romantic partners. A 17-year-old boy reported choosing not to disclose his TB diagnosis to his partner:

*"My girlfriend is not suspicious; I didn't tell her about my TB because she'll never want to see me again. She won't see it in the same way that others do. [. . .] She'll leave, I know her. [. . .] I don't want my friends to know, all of them because they are silly. That's why I just stayed at home. [. . .] I only told my best friend that I have chest problems and did not tell him about TB because he was going to tell them."*

Together, the loss in connectedness, (self-) isolation and fear of stigma, robbed many of the adolescents in our study the opportunity to form meaningful relationships with others.

Although adolescents should receive much of their support and both emotional and physical safety from household/family members, broader socio-economic and political contexts often shaped these relationships and the care they receive. For example, physical safety was a major concern for some participants. Crime, gangsterism, and violence were cited as an everyday reality. A 17-year-old boy explained that this reality led him to question whether he could do more with his life:

*"I tell myself I'm not going to make it, because all the children my age there by us, we die young. We're not even gangsters and we die, and that's why I have all these dreams. I have ambition, I have all that, but like [I question myself] what's the use of all that? Like, you work so hard for all that, knowing that someone can just end your life now."*

The 17-year-old boy went on to say that his father had once been a gangster, and although others expected he would end up a gangster himself, he was motivated to prove them wrong. Similarly, emotional safety was not a given for all participants. Some participants were entrusted with the responsibility of looking after themselves and their siblings. Others, like a 16-year-old boy, recalled hostile relationships with their parents, in which tensions around safety and care were in constant flux:

*"Most of the time my mother would leave early [on a Friday] because [. . .] she doesn't want to make food on weekends [. . .] Saturday and Friday, she's drinking. Sometimes she will sleep out. [. . .] There were many times when we were younger [. . .] we had to go look where she is. And we have to tell her she must come [home] and make food"*

In some families, underlying socioeconomic constraint and weak social networks precipitated tensions between adolescents and the rest of the family and were exacerbated following their TB diagnosis. One 19-year-old man recounted how after finding out about his TB diagnosis, his uncle, with whom he had been living, threatened him:

"*My uncle told me [. . .] he will look after me and give me food for this week but next week if I'm going to be there the Friday still, he's going to send someone out to hurt me because he doesn't want me in this place [. . .] I feared for my life.*"

## Impact on schooling and employment wellbeing

Most participants were school-going adolescents who struggled to manage their treatment and schoolwork simultaneously. Factors such as TB treatment, its side-effects, and stigmatization at schools and elsewhere, interfered with their education and potential job opportunities. A 19-year-old man with drug-susceptible TB explained: "*you struggle to concentrate when you're dosing which means your marks and things don't look that strong.*" Participants reported that they did not attend school for a period following their TB diagnosis. Some participants only stayed away from school for the first two weeks of treatment, while others reported staying away from school for three or more months, sometimes missing an entire term of school.

Beyond struggling to stay abreast of schoolwork due to TB and/or its treatment, some participants pointed to the apathetic or adversarial stance schools or teachers had towards people affected by TB. A 15-year-old girl with MDR-TB said:

"*At school, they don't care [. . .]. They did not send work for a long time. They failed to provide the homework so that I could study for the exams. They told me at school that I would have to repeat Grade 9.*"

Other participants' schoolwork was not as severely affected by TB and its treatment. These participants described how their school had accommodated their TB care needs and facilitated them continuing schoolwork at home. Another 15-year-old girl on MDR-TB treatment described how "*when it was time for the exams, I went to school and wrote in the library.*"

For a few participants, TB and its treatment occurred at critical periods in their schooling careers. An 18-year-old man complained that he was unable to "*finish Grade 12*", which is the final year of secondary schooling, delaying his ability to enter the workforce and earn a salary. Similarly, a 19-year-old woman who had struggled with her drug-susceptible TB diagnosis and treatment said the treatment left her "*vomiting [and] unable to eat, sleep, or do anything. So, I was unable to pursue my dreams [of] becoming a doctor.*" Another 19-year-old woman, similarly, explained that TB had disrupted her first year at college, negatively impacting her studies, which she felt endangered her future work opportunities. Far from being passive in her decision around college, however, she said she had purposefully decided to halt her studies as she felt it was better to prioritise her health:

"*I just felt that my health comes first [. . .] I will go back to school, but I must first finish my treatment.*"

## Impact on agency and resilience

Adolescents in this study reported limited agency due to being diagnosed with TB. Some participants described how their parents, other household members, teachers at school or, health workers made decisions on their behalf, including whom to disclose their TB diagnosis. Adolescents complained that their parents disclosed their TB diagnoses without their consent or knowledge. A 15-year-old girl who had MDR-TB explained: "*My aunts and uncles were visiting one weekend, and my mother told them about my TB diagnosis because she was afraid that they*

*will discriminate me."* For some participants, what their caregivers believed was supportive behaviour had become an overreach, effectively eradicating their agency and privacy. A 20-year-old woman described feeling like she was being treated *"like a new-born baby."* She went on to say that her family had violated her trust by reading her diary in which she wrote poetry about her experiences of MDR-TB.

Despite their limited decision-making power, adolescents still found ways to exercise agency. For example, a 16-year-old boy described delaying seeking care until his symptoms were debilitating, when he *"saw there's a lot of things I couldn't do.*" The same participant exercised his agency by skipping treatment administration when it was inconvenient, or when he was afraid of the social impact. He said, "*I have been missing my treatment a lot recently [. . .] because I am starting to go out with my friends.*"

Resilience in the face of TB and its treatment was surprisingly commonplace among our participants. Some participants pointed to the support they received from family, friends and/ or health workers to help them overcome their TB disease. As a 15-year-old girl on MDR-TB treatment explained,

> *"When I first found out that I have TB, I was angry. Because I remembered that my dad died of TB, and I thought I was going to die as well. But now I am better because nurses motivated me to take my medication."*

Despite the many psychosocial and economic challenges adolescents experienced, most exhibited resilience, not only in completing their treatment but also in expressing a desire to pursue their dreams. A 16-year-old boy who had drug-susceptible TB shared that he would like to write novels, start a business and then a family:

> *"For the next 6 month [my goal] is to get a laptop and start applying for a bursary at the end of my grade 11 year. [. . .]. After school I would like to study business and then establish myself and after that I would like to find a romantic partner."*

This detailed account illustrates how earnestly the participant has thought about and planned to achieve his goals. Another participant, a 12-year-old girl, spoke about how her drug-susceptible TB treatment journey had inspired her to pursue becoming a doctor. She said:

> *"When I went to the doctors, I could see how hard they work to make the people healthy and give people counselling, that's why I want to be a doctor."*

## Discussion

This study sought to understand adolescents' experiences of TB (both drug-susceptible TB-and MDR-TB) in the context of their psychosocial, economic and household contexts. We found that TB negatively impacts adolescents' everyday functioning, with some adolescents ceasing activities outside the household. Generally, adolescents experienced distress, isolation and tension in their romantic, platonic and familial relationships, preventing them from forming meaningful connections with others. As expected, environmental circumstances like unemployment and generalised insecurity were common among our participants and their families, hindering their ability to exercise their agency and make health-conscious decisions. TB disrupted adolescents schooling, raising concerns about the long-term impact on their financial freedom.

TB disease exacerbated these underlying economic constraints, forcing some participants and/or their families into debt or impoverishment. Additionally, many participants lived in areas with high crime rates, where gangsterism and violence were rife. Adolescents described their communities as undesirable places to live where maintaining a positive outlook on life, and forging a better future for themselves and their families seems near impossible.

Similarly to our work, Atkins et al.'s, [29] review revealed that TB has manifold socioeconomic impacts on children, adolescents and their households, including financial, stigma, education, and household functioning. Other research from India suggests that TB's impact on household income and financial stability can lead to cycles of poverty that can last generations [30]. Research among caregivers of children and adolescents affected by MDR-TB in South Africa suggested that high health-related costs negatively impacted household financial security, decreased household income and caused psychological distress [15]. Another study conducted in South Africa found that almost 30% of people treated for TB experienced catastrophic costs (costs totalling $\geq$20% of annual household income), due to the treatment process (including travel and diagnosis) and loss of income generation [31]. Other research suggests these costs are greater among people affected by MDR-TB and those co-infected with TB-HIV, than those only affected by drug-susceptible TB [14, 32, 33]. Another way in which adolescents' described TB negatively impacting their financial security was in the form of disruptions to their schooling and education. Adolescents reported receiving varied support from their school, with some receiving schoolwork while at home and others being prevented from attending school, completing schoolwork or even writing exams. Adolescents perceived delays to graduating from high school or beginning tertiary education as disrupting employment and income opportunities for themselves and their families. Similar research conducted in China showed that TB had a substantial impact on adolescents' schooling and career choices [34].

As other research has shown, we found that TB disease's negative impact on adolescents' mental health and social lives can disrupt their ability and/or willingness to adhere to treatment (Leddy., 2022). Similar to findings in other research, adolescents in our study reported feeling depressed, despondent, and at times suicidal—especially those with MDR-TB [35, 36]. A more generalised concern with stigma appeared pervasive amongst our participants. Participants described instances of enacted, perceived and internalised stigma. Other studies have reported how stigma among adolescents can negatively impact treatment access, adherence and completion [37–39].

This study makes an important contribution in that it is among the first to show the impact TB has on adolescents' intimate and romantic relationships—an otherwise underrepresented area of research [5, 40]. Adolescents in our study were especially distressed by the negative impact TB had on romantic and sexual relationships, reporting that they had lost self-confidence and refused to engage with the opposite sex. Additionally, a few adolescents reported wanting to refrain from potentially exposing their partners to TB, even after they had been on treatment for several weeks and were no longer infectious. This highlights the need for further counselling and education on infection, prevention and control among adolescents. While overall stigma-reduction efforts among adolescents and children are lacking [41], a recent scoping review suggests that individual-level counselling and social support interventions must be combined with information-based community-level interventions to effectively address stigma within the TB care cascade [42]. Although adolescents in our study reported receiving support from household/family members, research from South Africa suggests that household members likely experience similar levels of stigma, further underscoring the need for stigma-reduction interventions to target households, communities and health systems processes [43, 44].

TB patients recently diagnosed with TB in South Africa, can apply for a temporary government-funded disability grant, but only those who meet the means test criteria will receive the grant, leaving many without assistance [45]. People (>18-years-old) and caregivers applying on their child's behalf, with TB can receive the grant for six to twelve months, after which their eligibility is re-evaluated, not taking their broader socio-economic conditions into account [46]. Adolescents in this study reported that the SASSA grant helped them contribute to the household income and offset treatment-related costs. Research conducted in Cape Town found that social protection programmes have the potential to help households build resilience by lowering vulnerability and ameliorate the financial costs related to TB care [47]. Future research should explore the potential of social protection programmes provided to adolescents and their families, that goes beyond socio-economic support.

This study provides further impetus to the emerging understanding of adolescents' complex TB experiences. Specifically, it provides insights on how TB disrupts educational and potentially income-generating opportunities and how TB-related stigma can disrupt social relations, particularly, romantic partnerships. The study advocates for further research into the psychosocial factors that might contribute to ongoing financial, social and psychological impairments following TB episodes among adolescents. This study was limited to adolescents living in the Western Cape of South Africa, and findings cannot be completely generalised to the larger population. Strengths of this study include having multiple interactions with adolescents over several months, allowing researchers to build rapport with participants and refine interview questions. Additionally, the data underwent multiple revisions before being iteratively analysed.

TB had a substantial impact on the lives of adolescents in our study. Existing TB programmes focus on drug therapy with little consideration for holistic care. Until these more complex psychosocial and socioeconomic issues are addressed, it is unlikely that TB control efforts will be achieved in this population.

## Author Contributions

**Conceptualization:** Dillon T. Wademan, Mfundo Mlomzale, Arlene J. Marthinus, James A. Seddon, Graeme Hoddinott.

**Data curation:** Dillon T. Wademan, Mfundo Mlomzale, Arlene J. Marthinus.

**Formal analysis:** Dillon T. Wademan, Mfundo Mlomzale, Arlene J. Marthinus.

**Funding acquisition:** James A. Seddon, Graeme Hoddinott.

**Investigation:** Dillon T. Wademan, Stephanie Jacobs, Khanyisa Mcimeli, Klassina Zimri.

**Methodology:** Dillon T. Wademan, Mfundo Mlomzale, Arlene J. Marthinus.

**Project administration:** Dillon T. Wademan, Klassina Zimri.

**Resources:** Dillon T. Wademan.

**Software:** Dillon T. Wademan.

**Supervision:** James A. Seddon, Graeme Hoddinott.

**Validation:** Dillon T. Wademan, Mfundo Mlomzale.

**Visualization:** Mfundo Mlomzale, Graeme Hoddinott.

**Writing – original draft:** Dillon T. Wademan, Mfundo Mlomzale.

**Writing – review & editing:** Dillon T. Wademan, Mfundo Mlomzale, Arlene J. Marthinus, Klassina Zimri, James A. Seddon, Graeme Hoddinott.

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
