## [Decision Letter · Decision Letter 0]

11 Jun 2024

PGPH-D-24-00902

Psychosocial experiences of adolescents with tuberculosis in Cape Town

Dear Dr. Mlomzale,

Thank you for submitting your manuscript to PLOS Global Public Health. After careful consideration, we feel that it has merit but does not fully meet PLOS Global Public Health’s publication criteria as it currently stands. Therefore, we invite you to submit a revised version of the manuscript that addresses the points raised during the review process.

We look forward to receiving your revised manuscript.

Kind regards,

Sualeha Siddiq Shekhani

Academic Editor

Journal Requirements:

Additional Editor Comments (if provided):

Academic Editor Comments:

-The manuscript should be revised for punctuation and formatting errors.

-The beginning of the Discussion section can be made better. The first paragraph for example could offer a summary of the findings, and also explain how this study is different from the rest of the studies conducted on a similar subject.

Reviewers' comments:

Reviewer's Responses to Questions

**Comments to the Author**

1. Does this manuscript meet PLOS Global Public Health’s publication criteria? Is the manuscript technically sound, and do the data support the conclusions? The manuscript must describe methodologically and ethically rigorous research with conclusions that are appropriately drawn based on the data presented.

Reviewer #1: Yes

Reviewer #2: Yes

2. Has the statistical analysis been performed appropriately and rigorously?

Reviewer #1: I don't know

Reviewer #2: N/A

3. Have the authors made all data underlying the findings in their manuscript fully available (please refer to the Data Availability Statement at the start of the manuscript PDF file)?

Reviewer #1: No

Reviewer #2: Yes

4. Is the manuscript presented in an intelligible fashion and written in standard English?

Reviewer #1: Yes

Reviewer #2: Yes

5. Review Comments to the Author

Reviewer #1: Congratulations to the authors for drafting the manuscript.

Please address the queries below

1.Please summarize the study findings as a table at the end of the results section

2.Please avail the code book

3.Please avail a duly completed COREQ checklist as a supplementary

Reviewer #2: An intriguing article that explores adolescents experiences of TB in Cape Town.

I have two minor edits:

1. Line 235 - You have 'A 17-year-old you man'. You need to edit this line for correctness and clarity.

2. Line 240 - You have 'Many participants reported and isolating themselves from'. You need to edit this line for correctness and clarity.

6. PLOS authors have the option to publish the peer review history of their article (what does this mean?). If published, this will include your full peer review and any attached files.

**Do you want your identity to be public for this peer review?** For information about this choice, including consent withdrawal, please see our Privacy Policy.

Reviewer #1: **Yes: **Dr. Angela Nyangore Migowa

Reviewer #2: No

---

## [Decision Letter · Decision Letter 1]

9 Aug 2024

Psychosocial experiences of adolescents with tuberculosis in Cape Town

PGPH-D-24-00902R1

Dear Dr.Mlomzale 

We are pleased to inform you that your manuscript 'Psychosocial experiences of adolescents with tuberculosis in Cape Town' has been provisionally accepted for publication in PLOS Global Public Health.

Best regards,

Sualeha Siddiq Shekhani

Academic Editor

Reviewer Comments (if any, and for reference):

Reviewer's Responses to Questions

**Comments to the Author**

If the authors have adequately addressed your comments raised in a previous round of review and you feel that this manuscript is now acceptable for publication, you may indicate that here to bypass the “Comments to the Author” section, enter your conflict of interest statement in the “Confidential to Editor” section, and submit your "Accept" recommendation.

Reviewer #2: All comments have been addressed